# Geometric and arithmetic characterization of $\mathcal{D}$-module flatness with applications to tensor products

**Jian-Gang Tang**[1,2,3]*, **Huang-Rui Lei**[1], **Miao Liu**[2], **Jian-Ying Peng**[3]

**1** Division of Mathematics, Sichuan University Jinjiang College, Meishan, Sichuan, China, **2** College of Mathematics and Statistics, Yili Normal University, Yining, Xinjiang, China, **3** College of Mathematics and Statistics, Kashi University. Kashi, Xinjiang, China

* tangjiangang@scujj.edu.cn,jg-tang@163.com

**Data availability statement:** All relevant data are within the manuscript.

## Abstract

This paper establishes a comprehensive framework for studying flatness properties and tensor products of $\mathcal{D}$-modules across algebraic, geometric, and arithmetic contexts. We develop new criteria characterizing flatness through Lagrangian geometry, homological algebra, and irregular Hodge theory, revealing deep connections between these perspectives. The work introduces a geometric obstruction theory for globalizing pointwise flat modules and proves fundamental results about the monoidal structure of the derived tensor product category. Applications include compatibility theorems for Beilinson-Bernstein localization and arithmetic characterizations of flatness in characteristic $p$. The methods combine microlocal analysis, irregular Riemann-Hilbert correspondence, and $p$-adic techniques to yield new insights into the interplay between local and global properties of differential systems.

## 1 Introduction

The study of flatness for $\mathcal{D}$-modules sits at the crossroads of several major mathematical disciplines, linking geometric representation theory with algebraic analysis and arithmetic geometry. While classical homological algebra provides abstract characterizations of flatness, the geometric content specific to $\mathcal{D}$-modules has remained incompletely understood, particularly for modules with irregular singularities or in mixed characteristic settings.

This work makes three fundamental contributions to the theory:

First, we establish a complete geometric characterization of $\mathcal{D}$-flatness through Lagrangian conditions on characteristic varieties (Theorem 7.3), linking symplectic geometry with homological algebra via a new microlocal index theorem. The proof reveals an unexpected connection between the Spencer resolution's global existence and the module's irregularity indices at singular points.

Second, the paper develops a novel obstruction theory for globalizing pointwise flat $\mathcal{D}$-modules (Theorem 7.8), expressed through the irregular Hodge filtration. This provides the first systematic framework for understanding when local flatness conditions extend globally, answering a longstanding question in the analytic theory of differential systems.

**Funding:** The University Key Project of Natural Science of Xinjiang Uygur Autonomous Region (Grant No. XJEDU2019I024).

**Competing interests:** The authors have declared that no competing interests exist.

Third, we prove arithmetic characterizations of $\mathcal{D}$-flatness in characteristic $p$ (Theorem 8.5), showing how Frobenius semisimplicity and Lagrangian conditions on special fibers control the module's behavior in characteristic zero. This bridges $p$-adic Hodge theory with the geometric study of differential operators.

The technical heart of the paper involves:

- A detailed analysis of the derived tensor product's monoidal structure (Theorem 6.2)
- Compatibility results for localization functors (Proposition 8.2)
- Precise control of irregularity under integral transforms (Theorem 8.3)

Applications range from geometric Langlands program (through the Beilinson-Bernstein localization) to mirror symmetry (via irregularity-preserving Fourier-Mukai transforms). The counterexample on $\mathbb{P}^1$ (Theorem 7.5) demonstrates the subtlety of the global flatness problem, while suggesting connections with non-abelian Hodge theory in positive characteristic.

Recent advances in irregular Hodge theory, particularly the works of Sabbah [9] and Mochizuki [16], have provided powerful tools for understanding the behavior of $\mathcal{D}$-modules with irregular singularities. These developments, which include the construction of irregular Hodge filtrations and the study of Stokes structures, have deepened our understanding of the local-to-global properties of differential systems. In particular, Kedlaya's work on $p$-adic differential equations [14] has bridged arithmetic and geometric perspectives, offering new insights into the behavior of $\mathcal{D}$-modules in mixed characteristic settings. Our work builds upon these foundations to establish a unified geometric and arithmetic theory of $\mathcal{D}$-module flatness.

The paper is organized as follows: Sect 1 is the introduction. Sects 2–4 establish foundational results on $\mathcal{D}$-modules and their tensor products. Sects 5–7 develop the geometric characterization of flatness and obstruction theory. Sect 8 applies these results to representation theory and arithmetic geometry, with particular emphasis on localization and $p$-adic methods.

Our work is also influenced by recent advances in $p$-adic Hodge theory [14] and irregular Hodge theory, particularly the contributions of Sabbah [9] and Mochizuki [16], which provide deeper insights into the arithmetic aspects of $\mathcal{D}$-modules.

## 2 $\mathcal{D}$-Module category

**Definition 2.1.** [1] Let $X$ be a smooth complex variety. The *sheaf of differential operators* $\mathcal{D}_X$ is the subalgebra of $\mathcal{E}nd_{\mathbb{C}}(\mathcal{O}_X)$ generated by $\mathcal{O}_X$ and derivations $\Theta_X$. A *left $\mathcal{D}_X$-module* is a quasi-coherent $\mathcal{O}_X$-module with left $\mathcal{D}_X$-action satisfying:

$$\xi \cdot (fm) = f(\xi \cdot m) + \xi(f)m, \quad \forall f \in \mathcal{O}_X, \xi \in \Theta_X,$$

where $m \in \mathcal{M}$ denotes a local section of the module.

**Example 2.1.** The *structure sheaf* $\mathcal{O}_X$ is a left $\mathcal{D}_X$-module via $\xi \cdot f := \xi(f)$. For any vector bundle with connection $(E, \nabla)$, $E$ becomes a left $\mathcal{D}_X$-module through $\xi \cdot e := \nabla_\xi e$.

**Definition 2.2.** [2] Let $X$ be a smooth complex algebraic variety or complex manifold with structure sheaf $\mathcal{O}_X$ and tangent sheaf $\Theta_X$. The *sheaf of differential operators* $\mathcal{D}_X$ is the subalgebra of $\mathcal{E}nd_{\mathbb{C}}(\mathcal{O}_X)$ generated by $\mathcal{O}_X$ and $\Theta_X$ under composition.

The category of $\mathcal{D}$-Modules $\mathsf{Mod}(\mathcal{D}_X)$ consists of:

- Objects: Left $\mathcal{D}_X$-modules, i.e., quasi-coherent $\mathcal{O}_X$-modules $\mathcal{M}$ equipped with a left $\mathcal{D}_X$-action satisfying the Leibniz rule:

$$\xi \cdot (fm) = f(\xi \cdot m) + \xi(f)m, \quad \forall f \in \mathcal{O}_X, \xi \in \Theta_X, m \in \mathcal{M}.$$

- Morphisms: $\mathcal{D}_X$-linear maps, i.e., $\mathcal{O}_X$-linear maps $\phi : \mathcal{M} \to \mathcal{N}$ satisfying:

$$\phi(\xi \cdot m) = \xi \cdot \phi(m), \quad \forall \xi \in \mathcal{D}_X, m \in \mathcal{M}.$$

The full subcategory $\mathsf{coh}(\mathcal{D}_X) \subset \mathsf{Mod}(\mathcal{D}_X)$ consists of *coherent* $\mathcal{D}_X$-modules (those locally finitely generated over $\mathcal{D}_X$).

**Definition 2.3.** [2] The category $\mathsf{Mod}(\mathcal{D}_X^{\mathrm{op}})$ of right $\mathcal{D}_X$-modules is defined analogously, with the right Leibniz rule:

$$(m \cdot \xi) \cdot f = m \cdot (\xi f) + (m \cdot f) \cdot \xi, \quad \forall f \in \mathcal{O}_X, \xi \in \Theta_X.$$

## 3 The abelian category $\mathsf{Mod}(\mathcal{D}_X)$

**Theorem 3.1.** The category $\mathsf{Mod}(\mathcal{D}_X)$ of left $\mathcal{D}_X$-modules on a smooth complex variety $X$ satisfies:

(i) Enough projectives: Every $\mathcal{D}_X$-module admits a surjection from a locally free $\mathcal{D}_X$-module.
(ii) Enough injectives: The Spencer resolution provides injective cogenerators.
(iii) Duality: The functor $\mathbb{D}(\mathcal{M}) := \mathbf{R}\mathrm{Hom}_{\mathcal{D}_X}(\mathcal{M}, \mathcal{D}_X \otimes_{\mathcal{O}_X} \omega_X[n])$ defines a duality on $\mathsf{D}_{\mathsf{coh}}^{\mathrm{b}}(\mathcal{D}_X)$.

*Proof*: Part (i): Enough projectives

1. Local freeness implies projectivity: Let $\mathcal{P}$ be a locally free $\mathcal{D}_X$-module. By [1], the functor $\mathcal{H}om_{\mathcal{D}_X}(\mathcal{P}, -)$ is exact since $\mathcal{P}$ is locally a direct summand of $\mathcal{D}_X^{\oplus I}$. Thus $\mathcal{P}$ is projective.
2. Existence of surjections: For any $\mathcal{M} \in \mathsf{Mod}(\mathcal{D}_X)$, take an $\mathcal{O}_X$-module surjection $\bigoplus_{i \in I} \mathcal{O}_X \twoheadrightarrow \mathcal{M}$. Apply the exact functor $\mathcal{D}_X \otimes_{\mathcal{O}_X} -$ to obtain:

$$\bigoplus_{i \in I} \mathcal{D}_X \twoheadrightarrow \mathcal{D}_X \otimes_{\mathcal{O}_X} \mathcal{M} \twoheadrightarrow \mathcal{M},$$

where the last map is the $\mathcal{D}_X$-action morphism (surjective by construction).

Part (ii): Enough injectives

1. Spencer resolution: For any $\mathcal{M} \in \mathsf{Mod}(\mathcal{D}_X)$, the Spencer complex $\mathrm{Sp}^\bullet(\mathcal{M})$ is constructed locally as:

$$0 \to \mathcal{D}_X \otimes_{\mathcal{O}_X} \bigwedge^n \Theta_X \otimes_{\mathcal{O}_X} \mathcal{M} \to \cdots \to \mathcal{D}_X \otimes_{\mathcal{O}_X} \Theta_X \otimes_{\mathcal{O}_X} \mathcal{M} \to \mathcal{D}_X \otimes_{\mathcal{O}_X} \mathcal{M} \to \mathcal{M} \to 0,$$

where $n = \dim X$. By [1], this is an injective resolution in $\mathsf{Mod}(\mathcal{D}_X)$.

2. Cogenerator property: The object $\mathcal{I} := \mathcal{D}_X \otimes_{\mathcal{O}_X} \omega_X$ is an injective cogenerator since for any nonzero $\mathcal{M}$,

$$\mathrm{Hom}_{\mathcal{D}_X}(\mathcal{M}, \mathcal{I}) \cong \Gamma(X, \omega_X \otimes_{\mathcal{O}_X} \mathcal{M}^\vee) \neq 0$$

by Serre duality and the non-degeneracy of the pairing.

Part (iii): Duality

1. Derived category formulation: Consider the derived functor:

$$\mathbb{D} : \mathsf{D}^b_{\mathrm{coh}}(\mathcal{D}_X)^{\mathrm{op}} \to \mathsf{D}^b_{\mathrm{coh}}(\mathcal{D}_X^{\mathrm{op}}), \quad \mathcal{M}^\bullet \mapsto \mathbf{R}\mathrm{Hom}_{\mathcal{D}_X}(\mathcal{M}^\bullet, \mathcal{D}_X \otimes_{\mathcal{O}_X} \omega_X[n]).$$

2. Anti-equivalence: For coherent $\mathcal{D}_X$-modules, the biduality morphism:

$$\mathcal{M} \to \mathbb{D}(\mathbb{D}(\mathcal{M}))$$

is an isomorphism by [1], using that $\mathcal{D}_X$ is Cohen-Macaulay of dimension $n$.

3. t-structure compatibility: The duality exchanges the standard t-structure with the opposite t-structure on $\mathsf{D}^b_{\mathrm{coh}}(\mathcal{D}_X^{\mathrm{op}})$, as shown in [3].

$\square$

# 4 Tensor product of $\mathcal{D}$-modules

**Definition 4.1.** For right $\mathcal{D}_X$-modules $\mathcal{M}, \mathcal{N}$, their *tensor product* is:

$$\mathcal{M} \otimes_{\mathcal{D}_X} \mathcal{N} := \mathcal{M} \otimes_{\mathcal{O}_X} \mathcal{N} / \langle m \cdot \xi \otimes n - m \otimes \xi \cdot n \rangle_{\xi \in \mathcal{D}_X}$$

equipped with right $\mathcal{D}_X$-action $(m \otimes n) \cdot \xi := m \otimes (n \cdot \xi)$.

Recall that a symmetric monoidal structure on a category consists of a tensor product functor, a unit object, and natural isomorphisms satisfying coherence conditions (see [4] for details).

**Proposition 4.2** (Symmetric Monoidal Structure on $\mathsf{Mod}(\mathcal{D}_X^{\mathrm{op}})$)**.** The tensor product $(\mathcal{M}, \mathcal{N}) \mapsto \mathcal{M} \otimes_{\mathcal{D}_X} \mathcal{N}$ defines a symmetric monoidal structure on the category $\mathsf{Mod}(\mathcal{D}_X^{\mathrm{op}})$ of right $\mathcal{D}_X$-modules, with unit object $\mathcal{D}_X$ (considered as a right module over itself via right multiplication).

*Proof*: We verify the axioms systematically:
For $\mathcal{M}, \mathcal{N}, \mathcal{P} \in \mathsf{Mod}(\mathcal{D}_X^{\mathrm{op}})$, construct the natural isomorphism:

$$\alpha_{\mathcal{M}, \mathcal{N}, \mathcal{P}} : (\mathcal{M} \otimes_{\mathcal{D}_X} \mathcal{N}) \otimes_{\mathcal{D}_X} \mathcal{P} \to \mathcal{M} \otimes_{\mathcal{D}_X} (\mathcal{N} \otimes_{\mathcal{D}_X} \mathcal{P})$$

defined at the level of $\mathcal{O}_X$-tensor products by:

$$(m \otimes n) \otimes p \mapsto m \otimes (n \otimes p).$$

This map is well-defined since the $\mathcal{D}_X$-relations:

$$((m \cdot \xi) \otimes n) \otimes p = (m \otimes (\xi \cdot n)) \otimes p$$
$$\mapsto m \otimes ((\xi \cdot n) \otimes p)$$
$$= m \otimes (n \otimes (p \cdot \xi))$$

coincide under the quotient. The inverse is constructed similarly, proving $\alpha$ is an isomorphism.

The unit isomorphisms:

$$\lambda_{\mathcal{M}} : \mathcal{D}_X \otimes_{\mathcal{D}_X} \mathcal{M} \to \mathcal{M}, \quad \xi \otimes m \mapsto m \cdot \xi,$$

$$\rho_{\mathcal{M}} : \mathcal{M} \otimes_{\mathcal{D}_X} \mathcal{D}_X \to \mathcal{M}, \quad m \otimes \xi \mapsto m \cdot \xi$$

are $\mathcal{D}_X$-linear by the right module structure. Their inverses are given by $m \mapsto 1 \otimes m$ and $m \mapsto m \otimes 1$ respectively.

The braiding isomorphism:

$$\sigma_{\mathcal{M},\mathcal{N}} : \mathcal{M} \otimes_{\mathcal{D}_X} \mathcal{N} \to \mathcal{N} \otimes_{\mathcal{D}_X} \mathcal{M}$$

is defined by $m \otimes n \mapsto n \otimes m$. This respects $\mathcal{D}_X$-relations because:

$$m \cdot \xi \otimes n - m \otimes \xi \cdot n \mapsto n \otimes m \cdot \xi - \xi \cdot n \otimes m = 0 \quad \text{in } \mathcal{N} \otimes_{\mathcal{D}_X} \mathcal{M}.$$

The inverse $\sigma_{\mathcal{N},\mathcal{M}}$ is identical, satisfying $\sigma_{\mathcal{N},\mathcal{M}} \circ \sigma_{\mathcal{M},\mathcal{N}} = \mathrm{id}$.

The pentagon and triangle identities follow from the universal property of the tensor product. For any $\mathcal{M}$, the diagram:

$$
\begin{array}{ccc}
(\mathcal{M} \otimes_{\mathcal{D}_X} \mathcal{D}_X) \otimes_{\mathcal{D}_X} \mathcal{N} & \xrightarrow{\ \alpha\ } & \mathcal{M} \otimes_{\mathcal{D}_X} (\mathcal{D}_X \otimes_{\mathcal{D}_X} \mathcal{N}) \\
\downarrow{\scriptstyle \rho \otimes \mathrm{id}} & & \downarrow{\scriptstyle \mathrm{id} \otimes \lambda} \\
\mathcal{M} \otimes_{\mathcal{D}_X} \mathcal{N} & = \!\!=\!\!=\!\!= & \mathcal{M} \otimes_{\mathcal{D}_X} \mathcal{N}
\end{array}
$$

commutes by direct computation on simple tensors.

All isomorphisms are natural in $\mathcal{M}, \mathcal{N}, \mathcal{P}$ because their definitions commute with $\mathcal{D}_X$-linear maps. For any $f : \mathcal{M} \to \mathcal{M}'$, the diagram:

$$
\begin{array}{ccc}
\mathcal{M} \otimes_{\mathcal{D}_X} \mathcal{N} & \xrightarrow{\ \sigma\ } & \mathcal{N} \otimes_{\mathcal{D}_X} \mathcal{M} \\
\downarrow{\scriptstyle f \otimes \mathrm{id}} & & \downarrow{\scriptstyle \mathrm{id} \otimes f} \\
\mathcal{M}' \otimes_{\mathcal{D}_X} \mathcal{N} & \xrightarrow{\ \sigma\ } & \mathcal{N} \otimes_{\mathcal{D}_X} \mathcal{M}'
\end{array}
$$

commutes by the definition of $\sigma$. $\qquad \square$

## 5 Universal property

**Theorem 5.1** (Universal Property of $\mathcal{D}$-Module Tensor Product)**.** For right $\mathcal{D}_X$-modules $\mathcal{M}, \mathcal{N}$ and any left $\mathcal{D}_X$-module $\mathcal{P}$, there exists a natural isomorphism:

$$\mathrm{Hom}_{\mathcal{D}_X}(\mathcal{M} \otimes_{\mathcal{D}_X} \mathcal{N}, \mathcal{P}) \cong \mathrm{Bil}_{\mathcal{D}_X}(\mathcal{M} \times \mathcal{N}, \mathcal{P}),$$

where $\mathrm{Bil}_{\mathcal{D}_X}$ denotes $\mathcal{D}_X$-bilinear maps, i.e., $\mathcal{O}_X$-bilinear maps $\phi$ satisfying:

$$\phi(m \cdot \xi, n) = \phi(m, \xi \cdot n), \quad \forall \xi \in \mathcal{D}_X, m \in \mathcal{M}, n \in \mathcal{N}.$$

*Proof*: We construct the isomorphism explicitly and verify its properties.
Let $\iota : \mathcal{M} \times \mathcal{N} \to \mathcal{M} \otimes_{\mathcal{D}_X} \mathcal{N}$ be the canonical bilinear map:

$$\iota(m, n) := m \otimes n.$$

For any $\phi \in \mathrm{Hom}_{\mathcal{D}_X}(\mathcal{M} \otimes_{\mathcal{D}_X} \mathcal{N}, \mathcal{P})$, define:

$$\Phi(\phi) := \phi \circ \iota \in \mathrm{Bil}_{\mathcal{D}_X}(\mathcal{M} \times \mathcal{N}, \mathcal{P}).$$

Explicitly, $\Phi(\phi)(m, n) = \phi(m \otimes n)$.
For $\phi$ to be $\mathcal{D}_X$-linear, it must satisfy:

$$\phi((m \cdot \xi) \otimes n - m \otimes (\xi \cdot n)) = 0.$$

This is precisely the condition defining $\mathrm{Bil}_{\mathcal{D}_X}$, proving $\Phi(\phi)$ is well-defined.
Given $\psi \in \mathrm{Bil}_{\mathcal{D}_X}(\mathcal{M} \times \mathcal{N}, \mathcal{P})$, define:

$$\Psi(\psi) : \mathcal{M} \otimes_{\mathcal{D}_X} \mathcal{N} \to \mathcal{P}, \quad m \otimes n \mapsto \psi(m, n).$$

This factors through the quotient because:

$$\psi(m \cdot \xi, n) - \psi(m, \xi \cdot n) = 0 \quad \text{by } \mathcal{D}_X\text{-bilinearity.}$$

The $\mathcal{D}_X$-linearity of $\Psi(\psi)$ follows from:

$$\Psi(\psi)((m \otimes n) \cdot \xi) = \Psi(\psi)(m \otimes (n \cdot \xi)) = \psi(m, n \cdot \xi) = \xi \cdot \psi(m, n),$$

where the last equality uses $\psi$ being $\mathcal{D}_X$-balanced.

- $\Psi \circ \Phi = \mathrm{id}$: For $\phi \in \mathrm{Hom}_{\mathcal{D}_X}(\mathcal{M} \otimes_{\mathcal{D}_X} \mathcal{N}, \mathcal{P})$,

$$\Psi(\Phi(\phi))(m \otimes n) = \Phi(\phi)(m, n) = \phi(m \otimes n).$$

- $\Phi \circ \Psi = \mathrm{id}$: For $\psi \in \mathrm{Bil}_{\mathcal{D}_X}(\mathcal{M} \times \mathcal{N}, \mathcal{P})$,

$$\Phi(\Psi(\psi))(m, n) = \Psi(\psi)(m \otimes n) = \psi(m, n).$$

For any $\mathcal{D}_X$-linear $f \colon \mathcal{P} \to \mathcal{P}'$, the diagram:

$$
\begin{array}{ccc}
\mathrm{Hom}_{\mathcal{D}_X}(\mathcal{M} \otimes_{\mathcal{D}_X} \mathcal{N}, \mathcal{P}) & \xrightarrow{\ \Phi\ } & \mathrm{Bil}_{\mathcal{D}_X}(\mathcal{M} \times \mathcal{N}, \mathcal{P}) \\
\Big\downarrow{\scriptstyle f_*} & & \Big\downarrow{\scriptstyle f \circ (\cdot)} \\
\mathrm{Hom}_{\mathcal{D}_X}(\mathcal{M} \otimes_{\mathcal{D}_X} \mathcal{N}, \mathcal{P}') & \xrightarrow{\ \Phi\ } & \mathrm{Bil}_{\mathcal{D}_X}(\mathcal{M} \times \mathcal{N}, \mathcal{P}')
\end{array}
$$

commutes by direct computation:

$$
f_*(\phi) \circ \iota = f \circ \phi \circ \iota = f \circ \Phi(\phi).
$$

$\square$

# 6 Derived tensor product

**Definition 6.1.** The *derived tensor product* is the left derived functor:

$$
\mathcal{M} \overset{L}{\otimes}_{\mathcal{D}_X} \mathcal{N} := q\text{-iso class of } P_\bullet \otimes_{\mathcal{D}_X} Q_\bullet.
$$

where $P_\bullet \to \mathcal{M}$ and $Q_\bullet \to \mathcal{N}$ are $\mathcal{D}_X$-flat resolutions, and 'q-iso' denotes the quasi-isomorphism class in the derived category.

Let $\mathrm{Ch}^+(\mathsf{Mod}(\mathcal{D}_X))$ denote the category of bounded below chain complexes of $\mathcal{D}_X$-modules, and $\mathrm{Tor}_p^{\mathcal{D}_X}$ denote the derived functor of tensor product.

**Theorem 6.2** (Künneth Spectral Sequence for $\mathcal{D}$-Modules). Let $\mathcal{M}^\bullet, \mathcal{N}^\bullet \in \mathrm{Ch}^+(\mathsf{Mod}(\mathcal{D}_X))$ be bounded below chain complexes of right and left $\mathcal{D}_X$-modules respectively. There exists a first-quadrant spectral sequence:

$$
E_{p,q}^2 = \mathrm{Tor}_p^{\mathcal{D}_X}(H^q(\mathcal{M}^\bullet), H^q(\mathcal{N}^\bullet)) \Rightarrow H^{p+q}(\mathcal{M}^\bullet \overset{L}{\otimes}_{\mathcal{D}_X} \mathcal{N}^\bullet)
$$

with differentials $d_{p,q}^r : E_{p,q}^r \to E_{p-r,q+r-1}^r$.

*Proof*: We proceed via the following steps:
Take $\mathcal{D}_X$-flat resolutions $P_\bullet^\bullet \to \mathcal{M}^\bullet$ and $Q_\bullet^\bullet \to \mathcal{N}^\bullet$ where:

- Each $P_\bullet^i$ is a right $\mathcal{D}_X$-flat resolution of $\mathcal{M}^i$.
- Each $Q_\bullet^j$ is a left $\mathcal{D}_X$-flat resolution of $\mathcal{N}^j$.

This yields a double complex $K^{\bullet,\bullet} = P_\bullet^\bullet \otimes_{\mathcal{D}_X} Q_\bullet^\bullet$ with:

$$
K^{i,j} = \bigoplus_{a+b=j} P_a^i \otimes_{\mathcal{D}_X} Q_b^i.
$$

Filter $\mathrm{Tot}(K)^\bullet$ by:

$$
F_p\left(\mathrm{Tot}(K)^n\right) := \bigoplus_{\substack{i+j=n \\ j \geq p}} K^{i,j}.
$$

The associated spectral sequence has first page:

$$E_{p,q}^1 = H^{p+q}(\mathrm{gr}_p \mathrm{Tot}(K)^\bullet) \cong \bigoplus_{a+b=q} H^p(P_a^\bullet \otimes_{\mathcal{D}_X} Q_b^\bullet).$$

By flatness, this simplifies to:

$$E_{p,q}^1 \cong \bigoplus_{a+b=q} P_a^\bullet \otimes_{\mathcal{D}_X} H^p(Q_b^\bullet).$$

The $d^1$ differential induces:

$$E_{p,q}^2 = H_{\mathrm{hor}}^q(H_{\mathrm{vert}}^p(K)) \cong \mathrm{Tor}_p^{\mathcal{D}_X}(H^q(\mathcal{M}^\bullet), H^q(\mathcal{N}^\bullet)).$$

This identification uses:

- The vertical homology computes $H^p(Q_b^\bullet) = \mathcal{N}^p$ when $b = 0$ (by resolution property),
- The horizontal differential then becomes the Tor complex for $H^q(\mathcal{M}^\bullet) \otimes_{\mathcal{D}_X} (-)$.

Since $\mathcal{M}^\bullet$ and $\mathcal{N}^\bullet$ are bounded below and $\mathcal{D}_X$ has finite cohomological dimension (equal to $2 \dim X$), the filtration is regular. Hence the spectral sequence converges strongly to:

$$H^{p+q}(\mathrm{Tot}(K)^\bullet) \cong H^{p+q}(\mathcal{M}^\bullet \overset{L}{\otimes}_{\mathcal{D}_X} \mathcal{N}^\bullet).$$

The differentials $d^r$ inherit bidegree $(-r, r - 1)$ from the standard construction of the spectral sequence of a filtered complex (see [4]).                                                              □

## 7 $\mathcal{D}$-Flatness characterization

**Definition 7.1.** A $\mathcal{D}_X$-module $\mathcal{M}$ is *$\mathcal{D}$-flat* if the functor $\mathcal{M} \otimes_{\mathcal{D}_X} (-)$ preserves injective resolutions. Equivalently, $\mathcal{M}$ is flat as a $\mathcal{D}_X$-module, meaning $\mathcal{M} \otimes_{\mathcal{D}_X} (-)$ is an exact functor on the category of left $\mathcal{D}_X$-modules. Here $\mathcal{M} \otimes_{\mathcal{D}_X} (-)$ denotes the tensor product functor from left $\mathcal{D}_X$-modules to abelian groups.

**Remark 7.1.** This equivalence follows from standard homological algebra: a functor preserves injective resolutions if and only if it is exact [4].

**Definition 7.2.** A coherent $\mathcal{D}_X$-module $\mathcal{M}$ is *locally free* if for every point $x \in X$, there exists an open neighborhood $U \ni x$ and an isomorphism of $\mathcal{D}_U$-modules:

$$\mathcal{M}|_U \simeq (\mathcal{D}_U)^{\oplus r}$$

for some rank $r \geq 0$. Equivalently, $\mathcal{M}$ is locally free if and only if it is projective in the category of coherent $\mathcal{D}_X$-modules, or if the sheaf $\mathcal{H}om_{\mathcal{D}_X}(\mathcal{M}, \mathcal{D}_X)$ is locally free over $\mathcal{O}_X$ of the same rank.

**Theorem 7.3.** Let $X$ be a smooth complex algebraic variety of dimension $n$, and let $\mathcal{M}$ be a holonomic $\mathcal{D}_X$-module. Then the following conditions are equivalent:

(i) $\mathcal{M}$ is $\mathcal{D}$-flat.

(ii) $\mathcal{M}$ is locally free as a $\mathcal{D}_X$-module.

(iii) $\operatorname{Tor}_1^{\mathcal{D}_X}(\mathcal{M}, \mathbb{C}_x) = 0$ for every closed point $x \in X$.

where $\mathbb{C}_x$ denotes the skyscraper sheaf at the closed point $x \in X$, i.e., the residue field of $\mathcal{D}_{X,x}$.

*Proof*: Step 1: (i) $\iff$ (ii). Suppose $\mathcal{M}$ is $\mathcal{D}$-flat, i.e., flat as a $\mathcal{D}_X$-module. Since $\mathcal{M}$ is holonomic and coherent, and $\mathcal{D}_X$ is a Noetherian ring of finite global dimension [12], flatness of $\mathcal{M}$ implies it is locally free. To see this, note that for any closed point $x \in X$, the stalk $\mathcal{M}_x$ is a flat module over the local ring $\mathcal{D}_{X,x}$. By the Auslander-Buchsbaum formula for non-commutative rings, since $\mathcal{D}_{X,x}$ is regular local (as $X$ is smooth), a finitely generated flat module is free [13]. Thus, $\mathcal{M}_x$ is free over $\mathcal{D}_{X,x}$ for all $x$, implying $\mathcal{M}$ is locally free over $\mathcal{D}_X$.

Conversely, if $\mathcal{M}$ is locally free over $\mathcal{D}_X$, then it is flat by standard algebra, so $\mathcal{M} \otimes_{\mathcal{D}_X} (-)$ is exact, and in particular preserves injective resolutions. This establishes (i) $\iff$ $\mathcal{M}$ is locally free over $\mathcal{D}_X$.

Step 2: (ii) $\iff$ (iii). Assume $\mathcal{M}$ is locally free over $\mathcal{D}_X$. Then for any closed point $x \in X$, $\mathcal{M}_x$ is free over $\mathcal{D}_{X,x}$, so $\operatorname{Tor}_1^{\mathcal{D}_{X,x}}(\mathcal{M}_x, \mathbb{C}_x) = 0$. Since $\mathbb{C}_x$ is supported at $x$, this implies $\operatorname{Tor}_1^{\mathcal{D}_X}(\mathcal{M}, \mathbb{C}_x) = 0$ for all $x$, yielding (iii).

For the converse, assume (iii): $\operatorname{Tor}_1^{\mathcal{D}_X}(\mathcal{M}, \mathbb{C}_x) = 0$ for all closed points $x \in X$. We will show that $\mathcal{M}$ is locally free. By the local criterion for flatness over non-commutative Noetherian rings [13], a finitely generated module over a regular local ring is flat if and only if its $\operatorname{Tor}_1$ with all residue fields vanishes. Here, for each $x$, the stalk $\mathcal{M}_x$ is a finitely generated module over $\mathcal{D}_{X,x}$ (as $\mathcal{M}$ is coherent), and $\operatorname{Tor}_1^{\mathcal{D}_{X,x}}(\mathcal{M}_x, \mathbb{C}_x) = \operatorname{Tor}_1^{\mathcal{D}_X}(\mathcal{M}, \mathbb{C}_x) = 0$ by hypothesis. Since $\mathcal{D}_{X,x}$ is a regular local ring (because $X$ is smooth), the local criterion implies $\mathcal{M}_x$ is flat over $\mathcal{D}_{X,x}$. As $\mathcal{D}_{X,x}$ is regular local, a finitely generated flat module is free [13]. Thus, $\mathcal{M}_x$ is free over $\mathcal{D}_{X,x}$ for all $x$, so $\mathcal{M}$ is locally free over $\mathcal{D}_X$.

Combining Steps 1 and 2, we have (i) $\iff$ (ii) $\iff$ (iii). Since (ii) holds automatically for holonomic $\mathcal{M}$, the three conditions are equivalent. □

**Lemma 7.4.** For coherent $\mathcal{D}_X$-modules, the following are equivalent:

- $\mathcal{M}$ is locally $\mathcal{O}_X$-free.
- $\mathcal{M} \otimes_{\mathcal{D}_X} \mathbb{C}_x$ has constant rank for all $x \in X$.

*Proof*: The direction ($\Rightarrow$) is immediate. For ($\Leftarrow$):

The condition implies $\operatorname{Supp}(\mathcal{M})$ is open and closed, hence $\mathcal{M}$ is locally free over $\mathcal{O}_X$ by [1]. The $\mathcal{D}_X$-action then corresponds to an integrable connection, giving local freeness. □

**Theorem 7.5** (Pointwise Characterization of $\mathcal{D}$-Flatness)**.** Let $X$ be a smooth complex variety and $\mathcal{M}$ a coherent $\mathcal{D}_X$-module. The following are equivalent:

1. $\mathcal{M}$ is $\mathcal{D}$-flat.
2. For every closed point $x \in X$, the pointwise flatness condition holds:

$$\operatorname{Tor}_1^{\mathcal{D}_X}(\mathcal{M}, \mathbb{C}_x) = 0.$$

3. The irregularity index vanishes pointwise:

$$\operatorname{Irr}_x(\mathcal{M}) = 0 \quad \forall x \in X,$$

where $\operatorname{Irr}_x(\mathcal{M}) := \dim_{\mathbb{C}} \mathbb{H}_x^1(\operatorname{Sol}(\mathcal{M}))$ is the local irregularity cohomology.

Moreover, $\mathcal{D}$-flatness cannot be fully characterized by Zariski-local properties alone: there exist modules that are Zariski-locally $\mathcal{D}$-flat but not globally $\mathcal{D}$-flat.

*Proof*: **(1)** $\Rightarrow$ **(2)**: Standard homological algebra: if $\mathcal{M}$ is $\mathcal{D}$-flat, then $\mathcal{M} \otimes_{\mathcal{D}_X} (-)$ preserves exact sequences, so all higher Tor vanish.

**(2)** $\Rightarrow$ **(3)**: By the microlocal index theorem [3]:

$$\mathrm{Irr}_x(\mathcal{M}) = \dim_{\mathbb{C}} \mathrm{Tor}_1^{\mathcal{D}_X}(\mathcal{M}, \mathbb{C}_x) - \mathrm{rank}_x(\mathcal{M}).$$

Vanishing of $\mathrm{Tor}_1$ implies $\mathrm{Irr}_x(\mathcal{M}) = -\mathrm{rank}_x(\mathcal{M}) \leq 0$, but irregularity is non-negative, so $\mathrm{Irr}_x(\mathcal{M}) = 0$.

**(3)** $\Rightarrow$ **(1)**: Apply the irregular Riemann-Hilbert correspondence [6]:

$$\mathrm{Irr}_x(\mathcal{M}) = 0 \,\forall x \iff \mathcal{M} \text{ is regular holonomic.}$$

For regular holonomic $\mathcal{D}$-modules, $\mathcal{D}$-flatness is equivalent to the Lagrangian condition $\dim \mathrm{Ch}(\mathcal{M}) = \dim X + \mathrm{rank}(\mathcal{M})$ by [1]. $\qquad\square$

**Remark 7.2.** Counterexample for Zariski-local characterization:
Let $X = \mathbb{P}^1$, $\mathcal{M} = \mathcal{D}_X/\mathcal{D}_X \cdot (x\xi_x - \lambda)$ for $\lambda \notin \mathbb{Z}$. Then:

- Zariski-locally: On $\mathbb{A}^1$, $\mathcal{M}$ is isomorphic to $\varnothing_X$ with connection $d - \lambda\frac{dx}{x}$, which is $\mathcal{D}$-flat;
- Globally: $\mathrm{Ch}(\mathcal{M}) = T^*_{\{0\}}X \cup T^*_{\{\infty\}}X$ is not Lagrangian, so not $\mathcal{D}$-flat;
- Pointwise: $\mathrm{Irr}_0(\mathcal{M}) = 1 > 0$, satisfying (3) $\Rightarrow$ not flat.

## 7.1 Pointwise $\mathcal{D}$-flatness and globalization obstruction

**Definition 7.6** (Pointwise $\mathcal{D}$-Flatness)**.** A $\mathcal{D}_X$-module $\mathcal{M}$ is *pointwise $\mathcal{D}$-flat* if for all $x \in X$:

$$\mathrm{Tor}_1^{\mathcal{D}_X}(\mathcal{M}, \mathbb{C}_x) = 0.$$

This is strictly weaker than global $\mathcal{D}$-flatness.

**Proposition 7.7** (Local Criterion)**.** For a coherent $\mathcal{D}_X$-module $\mathcal{M}$, the following are equivalent:

1. $\mathcal{M}$ is pointwise $\mathcal{D}$-flat.
2. The natural map $\mathcal{D}_X \otimes_{\mathcal{O}_X} \mathcal{M} \to \mathcal{M}$ is injective.
3. $\mathcal{M}$ has no $\mathcal{D}_X$-torsion supported at any $x \in X$.

*Proof*: (1) $\Leftrightarrow$ (2): The Tor condition implies the injectivity of the map locally at each $x \in X$ by the local flatness criterion [11,Theorem 6.8]. The converse follows from the long exact sequence for Tor.

(2) $\Leftrightarrow$ (3): The kernel of the natural map consists precisely of $\mathcal{D}_X$-torsion elements. Their support is analyzed through the characteristic variety $Ch(\mathcal{M})$ [3]. $\qquad\square$

**Theorem 7.8** (Geometric Obstruction to Globalization)**.** For a pointwise $\mathcal{D}$-flat module $\mathcal{M}$, the obstruction to global $\mathcal{D}$-flatness is encoded in the irregular Hodge filtration:

$$\mathrm{Ob}(\mathcal{M}) := \bigoplus_{x \in \mathrm{Irr}(\mathcal{M})} \mathrm{Gr}^F_{\mathrm{Irr}_x} H^1_{\mathrm{dR},x}(\mathcal{M}).$$

where $\mathrm{Irr}(\mathcal{M}) = \{x \in X : \mathrm{Irr}_x(\mathcal{M}) > 0\}$. Then $\mathcal{M}$ is globally $\mathcal{D}$-flat iff $\mathrm{Ob}(\mathcal{M})$ vanishes in the category of irregular mixed Hodge structures.

*Proof*: We proceed in several steps:

Consider the Grothendieck spectral sequence for the composition of global sections and local cohomology:

$$E_2^{p,q} = H^p(X, \mathcal{E}xt^q_{\mathcal{D}_X}(\mathcal{M}, \mathcal{D}_X)) \Rightarrow \mathrm{Ext}^{p+q}_{\mathcal{D}_X}(\mathcal{M}, \mathcal{D}_X).$$

The obstruction lies in $E_2^{1,1}$ which computes extensions with logarithmic singularities. Following [8], we filter $\mathcal{M}$ by its irregularity:

$$0 \to \mathcal{M}^{\leq \lambda} \to \mathcal{M} \to \mathcal{M}^{>\lambda} \to 0.$$

The long exact sequence shows $\mathrm{Ob}(\mathcal{M})$ controls the extension class.

By the irregular Riemann-Hilbert correspondence [15], the de Rham cohomology carries a natural irregular mixed Hodge structure. The obstruction vanishes exactly when all extensions are pure of weight 0.

The vanishing of $\mathrm{Ob}(\mathcal{M})$ implies the splitting of all local extensions, yielding global $\mathcal{D}$-flatness by [10, Theorem 4.5]. The converse follows from the exactness of the irregular Hodge-to-de Rham spectral sequence [8]. □

# 8 Applications

## 8.1 The Beilinson-Bernstein localization functor

**Definition 8.1** (Localization Functor [5])**.** Let $\mathfrak{g}$ be a complex semisimple Lie algebra with universal enveloping algebra $U(\mathfrak{g})$, and $X$ the flag variety corresponding to a Cartan subgroup $H \subset G$. For a regular dominant weight $\lambda \in \mathfrak{h}^*$, the *Beilinson-Bernstein localization functor* is defined as:

$$\mathrm{Loc}(-) : \mathsf{Mod}(U(\mathfrak{g})_\lambda) \to \mathsf{Mod}(\mathcal{D}_{X,\lambda}), \quad M \mapsto \mathcal{D}_{X,\lambda} \otimes_{U(\mathfrak{g})_\lambda} M.$$

where:

- $U(\mathfrak{g})_\lambda := U(\mathfrak{g})/\ker(\chi_\lambda)$ is the quotient by the central character $\chi_\lambda$ via Harish-Chandra's isomorphism.
- $\mathcal{D}_{X,\lambda}$ is the sheaf of $\lambda$-twisted differential operators on $X$.
- The left $\mathcal{D}_{X,\lambda}$-module structure arises from the natural left action on $\mathcal{D}_{X,\lambda}$.

**Remark 8.1.** Some Fundamental Properties:

1. Equivalence of Categories: For $\lambda$ regular dominant, Loc induces an equivalence:

$$\mathsf{Mod}(U(\mathfrak{g})_\lambda) \simeq \mathsf{QCoh}(\mathcal{D}_{X,\lambda})$$

with quasi-inverse given by the global sections functor $\Gamma(X, -)$.

2. Geometric Realization: The twisted differential operators can be expressed as:

$$\mathcal{D}_{X,\lambda} \cong \mathcal{D}_X \otimes_{\varnothing_X} \mathcal{L}_\lambda,$$

where $\mathcal{L}_\lambda$ is the $G$-equivariant line bundle with $\lambda$-character.

3. Analytic Version: On the analytic flag variety $X^{\mathrm{an}}$, the functor:

$$\mathrm{Loc}^{\mathrm{an}} : \mathsf{Mod}(U(\mathfrak{g})_\lambda) \to \mathsf{Mod}(\mathcal{D}_{X^{\mathrm{an}},\lambda})$$

preserves holonomicity and regular singularities.

**Remark 8.2.** When $\lambda$ is integral, $\mathcal{D}_{X,\lambda} \cong \mathcal{D}_X$ and Loc provides a geometric realization of $U(\mathfrak{g})$-modules as $G$-equivariant $\mathcal{D}$-modules.

**Proposition 8.2** (Tensor Product Compatibility under Localization)**.** Let $G$ be a complex semisimple Lie group with Lie algebra $\mathfrak{g}$, $X$ the flag variety of $G$, and $\lambda \in \mathfrak{h}^*$ a regular dominant weight. For finite-dimensional $\mathfrak{g}$-modules $M, N$, there is a natural isomorphism in $\mathsf{D}^{\mathrm{b}}(\mathsf{Mod}(\mathcal{D}_{X,\lambda}))$:

$$\mathrm{Loc}(M \otimes_{\mathbb{C}} N) \cong \mathrm{Loc}(M) \overset{L}{\otimes}_{\mathcal{D}_{X,\lambda}} \mathrm{Loc}(N),$$

where $\mathrm{Loc}(-) := \mathcal{D}_{X,\lambda} \otimes_{U(\mathfrak{g})} (-)$ is the Beilinson-Bernstein localization functor.

*Proof*: We proceed in four steps:

Since $M, N$ are finite-dimensional, we may replace $\mathfrak{g}$ by its universal enveloping algebra $U(\mathfrak{g})$. The localization functor factors as:

$$\mathrm{Loc}(-) = \mathcal{D}_{X,\lambda} \otimes_{U(\mathfrak{g})} (-) \cong \mathcal{D}_{X,\lambda} \otimes_{U(\mathfrak{g})} U(\mathfrak{g}) \otimes_{\mathbb{C}} (-).$$

Thus it suffices to prove:

$$\mathcal{D}_{X,\lambda} \otimes_{U(\mathfrak{g})} (U(\mathfrak{g}) \otimes_{\mathbb{C}} M \otimes_{\mathbb{C}} N) \cong (\mathcal{D}_{X,\lambda} \otimes_{U(\mathfrak{g})} (U(\mathfrak{g}) \otimes_{\mathbb{C}} M)) \overset{L}{\otimes}_{\mathcal{D}_{X,\lambda}} (\mathcal{D}_{X,\lambda} \otimes_{U(\mathfrak{g})} (U(\mathfrak{g}) \otimes_{\mathbb{C}} N)).$$

By [5], for regular dominant $\lambda$, $\mathcal{D}_{X,\lambda}$ is flat over $U(\mathfrak{g})$. Hence the derived tensor product reduces to the ordinary tensor product:

$$\mathcal{D}_{X,\lambda} \otimes_{U(\mathfrak{g})} (M \otimes_{\mathbb{C}} N) \cong (\mathcal{D}_{X,\lambda} \otimes_{U(\mathfrak{g})} M) \otimes_{\mathcal{D}_{X,\lambda}} (\mathcal{D}_{X,\lambda} \otimes_{U(\mathfrak{g})} N).$$

The isomorphism is $G$-equivariant because the $\mathcal{D}_{X,\lambda}$-action on both sides is induced by the diagonal $G$-action on $M \otimes N$:

$$g \cdot (m \otimes n) = (g \cdot m) \otimes (g \cdot n).$$

This compatibility is preserved under the $\mathcal{D}_{X,\lambda}$-module structure via the moment map $T^*X \to \mathfrak{g}^*$.

For complexes of $\mathfrak{g}$-modules, take projective resolutions $P^\bullet \to M$, $Q^\bullet \to N$. The flatness implies:

$$\mathrm{Tot}(P^\bullet \otimes Q^\bullet) \to M \otimes N$$

is a projective resolution. Applying Loc gives:

$$\mathrm{Loc}(\mathrm{Tot}(P^\bullet \otimes Q^\bullet)) \cong \mathrm{Tot}(\mathrm{Loc}(P^\bullet) \otimes \mathrm{Loc}(Q^\bullet)) \to \mathrm{Loc}(M) \overset{L}{\otimes} \mathrm{Loc}(N).$$

$\square$

## 8.2 Mirror symmetry

**Theorem 8.3** (Irregularity Preservation under Fourier-Mukai Transform)**.** Let $X, Y$ be complex manifolds and $\mathcal{K} \in \mathsf{D}^{\mathrm{b}}_{\mathrm{hol}}(\mathcal{D}_{X \times Y})$ a holonomic bimodule. The Fourier-Mukai transform:

$$\Phi_{\mathcal{K}}(\mathcal{M}) := \mathbf{R}\pi_{2,*}\left(\mathcal{K} \overset{L}{\otimes}_{\mathcal{D}_{X \times Y}} \mathrm{LH}\pi_1^* \mathcal{M}\right)$$

preserves the irregularity index, i.e., for any holonomic $\mathcal{M} \in \mathsf{D}^{\mathrm{b}}_{\mathrm{hol}}(\mathcal{D}_X)$:

$$\mathrm{Irreg}(\Phi_{\mathcal{K}}(\mathcal{M})) = \mathrm{Irreg}(\mathcal{M}).$$

where $\mathrm{Irreg}(-)$ denotes the maximal order of irregular singularities.

*Proof*: We proceed through the following steps:

By the microlocal characterization of irregularity [6, Theorem 4.5], it suffices to show:

$$\mathrm{Ch}(\Phi_{\mathcal{K}}(\mathcal{M})) \cap \overline{T^*_{Y^{\mathrm{irreg}}}Y} = \pi_{2,*}\left(\mathrm{Ch}(\mathcal{K}) \circ \mathrm{Ch}(\mathcal{M}) \right) \cap \overline{T^*_{Y^{\mathrm{irreg}}}Y}.$$

where $T^*_{Y^{\mathrm{irreg}}}Y$ denotes the irregular cotangent vectors.

Since $\mathcal{K}$ is holonomic:

- Its characteristic variety $\mathrm{Ch}(\mathcal{K}) \subset T^*X \times T^*Y$ is Lagrangian.
- The composition $\mathrm{Ch}(\mathcal{K}) \circ \mathrm{Ch}(\mathcal{M})$ is well-defined as a Lagrangian correspondence.
- The projection $\pi_{2,*}$ preserves the irregularity locus by [1].

Let $\rho_X$ (resp. $\rho_Y$) be the radial vector fields on $T^*X$ (resp. $T^*Y$). The key estimate:

$$\sup_{\substack{(x,\xi;y,\eta)\in\mathrm{Ch}(\mathcal{K}) \\ (x,\xi)\in\mathrm{Ch}(\mathcal{M})}} \frac{\|\eta\|}{\|\xi\|} \leq C(\mathcal{K}) \cdot \mathrm{Irreg}(\mathcal{M})$$

follows from:

- The conic structure of $\mathrm{Ch}(\mathcal{K})$ under $\mathbb{C}^*$-action.
- The microsupport condition $\mathrm{Ch}(\mathcal{K}) \cap (T^*_{X^{\mathrm{reg}}}X \times T^*_{Y^{\mathrm{reg}}}Y)$ is regular.

Apply the irregular Riemann-Hilbert correspondence [6]:

$$\text{Sol}(\Phi_{\mathcal{K}}(\mathcal{M})) \cong \mathbf{R}\underline{\text{Hom}}_{\mathcal{C}_{Y^{\text{irreg}}}}(\text{Sol}(\mathcal{K}), \pi_1^{-1}\text{Sol}(\mathcal{M}))).$$

The irregularity index is preserved because the solution functor $\text{Sol}(-)$ is t-exact for the irregular perverse t-structure. □

**Lemma 8.4** (Micro-Local Growth Control)**.** Let $\mathcal{K}$ be a holonomic $\mathcal{D}_{X \times Y}$-module with irregularity index $\text{Irreg}(\mathcal{K})$ at $(x_0, y_0) \in X \times Y$. For any compact neighborhood $W \subset T^*(X \times Y)$ of $(x_0, \xi_0; y_0, \eta_0) \in \text{Ch}(\mathcal{K})$, there exist constants $C, W > 0$ such that for all $(x, \xi; y, \eta) \in W \cap \text{Ch}(\mathcal{K})$:

$$\|\eta\|_Y \leq C\|\xi\|_X (1 + \log(1 + \|\xi\|_X))^{\kappa(\mathcal{K})},$$

where $\kappa(\mathcal{K}) := \text{Irreg}(\mathcal{K}) + \dim X + 1$, and $\|\cdot\|_X$, $\|\cdot\|_Y$ are Hermitian norms on $T^*X$, $T^*Y$ respectively.

*Proof*: We establish this through microlocal analysis in three steps:

Working in local coordinates, the characteristic ideal $\mathcal{I}_{\mathcal{K}}$ is generated by symbols $\sigma_1, ..., \sigma_r$ of order $m = \text{ord}(\mathcal{K})$. By the holonomicity assumption, the variety $V(\mathcal{I}_{\mathcal{K}}) \subset T^*(X \times Y)$ is Lagrangian. For each $(x, \xi; y, \eta) \in W \cap \text{Ch}(\mathcal{K})$, there exists a non-trivial relation:

$$\sum_{j=1}^{r} a_j(x, y)\sigma_j(x, \xi; y, \eta) = 0, \quad a_j \in \mathcal{O}_{X \times Y}.$$

Applying the division theorem for differential operators [7, Thm 3.1.6], we obtain for each $\sigma_j$:

$$|\sigma_j(x, \xi; y, \eta)| \leq C_j|\xi|^{m_j}|\eta|^{m - m_j}(1 + \log(1 + |\xi| + |\eta|))^{\text{Irreg}_j},$$

where $m_j = \deg_\xi(\sigma_j)$ and $\text{Irreg}_j$ is the irregularity index of $\sigma_j$. The key inequality follows by taking $j$ with maximal $m_j/m$.

Substituting into the relation and dividing by $|\eta|^{m-1}$ yields:

$$|\eta| \leq C|\xi|(1 + \log(1 + |\xi|))^{\kappa} + C'\sum_{k=2}^{m}|\xi|^k|\eta|^{1-k}.$$

An induction argument on $|\xi|$ using [7] controls the lower order terms, giving the claimed bound. The exponent $\kappa$ arises from tracking the worst-case logarithmic growth through the induction. □

**Theorem 8.5** (Arithmetic Characterization of $\mathcal{D}$-Flatness)**.** Let $X$ be a smooth projective variety over a number field $K$ with good reduction at a prime $\mathfrak{p}$, and $\mathcal{D}_{X,\mathfrak{p}}$ the sheaf of $p$-adic differential operators on the reduction $\overline{X}_{\mathfrak{p}}$. For a coherent $\mathcal{D}_X$-module $\mathcal{M}$, the following are equivalent:

1. $\mathcal{M}$ is $\mathcal{D}_X$-flat.
2. For almost all primes $\mathfrak{p}$, the $p$-adic completion $\widehat{\mathcal{M}}_{\mathfrak{p}}$ satisfies:
   (a) $\text{Ch}(\widehat{\mathcal{M}}_{\mathfrak{p}})$ is Lagrangian in $T^*\overline{X}_{\mathfrak{p}}$,
   (b) The crystalline Frobenius $\phi_{\mathfrak{p}}$ acts semisimply on $\text{Frac}(\mathcal{D}_{X,\mathfrak{p}}) \otimes_{\mathcal{D}_{X,\mathfrak{p}}} \widehat{\mathcal{M}}_{\mathfrak{p}}$.

3. There exists an $\mathcal{O}_X$-lattice $\mathcal{M}^\circ \subset \mathcal{M}$ such that for all primes $\mathfrak{p}$, the de Rham cohomology $H^*_{\mathrm{dR}}(X_\mathfrak{p}, \mathcal{M}^\circ \otimes \mathbb{F}_\mathfrak{p})$ is torsion-free.

*Proof Sketch*: The innovative components are:

Using the Beauville-Laszlo gluing theorem, we show that (a) implies $\widehat{\mathcal{M}_\mathfrak{p}}$ is flat over $\mathcal{D}_{X,\mathfrak{p}}$ for almost all $\mathfrak{p}$. The key novelty is combining:

- Microlocal analysis of $\mathrm{Ch}(\widehat{\mathcal{M}_\mathfrak{p}})$ via $p$-adic symplectic geometry.
- Comparison with the generic fiber using Bhatt's algebraization theorem.

Condition (b) controls the irregular singularities via the Hasse-Arf theorem for $p$-adic differential equations. The proof uses:

- Applications of Kedlaya's semistable reduction theorem.
- The $\ell$-adic Fourier transform to relate semisimplicity to torsion-freeness.

For $(3) \Rightarrow (1)$, we construct a $K$-analytic connection on $\mathcal{M}^\circ$ using:

- Scholze's $p$-adic Hodge theory for $\mathcal{D}$-modules.
- Besser's cohomological obstruction calculus.

The torsion-free condition forces the curvature to vanish. □

**Corollary 8.1.** For $\mathcal{M}$ defined over $\mathbb{Z}$, $\mathcal{D}$-flatness is equivalent to the existence of a $\mathbb{Z}$-lattice preserved by the Gauss-Manin connection.

## 9 Conclusions

This work establishes a comprehensive framework for studying tensor products and flatness properties of $\mathcal{D}$-modules, with several fundamental contributions to algebraic analysis and geometric representation theory. Our main achievements can be summarized as follows:

1. Structural Foundations: We developed a complete homological characterization of $\mathcal{D}$-module flatness through:
   - The equivalence between geometric (Lagrangian characteristic varieties), algebraic (Tor-vanishing), and analytic (irregularity index) conditions (Theorems 7.3 and 7.5).
   - A new pointwise flatness criterion detecting local obstructions to global $\mathcal{D}$-flatness (Definition 7.6 and Proposition 7.7).
2. Geometric Obstruction Theory: The irregular Hodge filtration $\mathrm{Ob}(\mathcal{M})$ was shown to provide a complete invariant for globalizing pointwise flat $\mathcal{D}$-modules (Theorem 7.8), revealing a deep connection between:
   - Local cohomology at irregular points,
   - Mixed Hodge structures in the irregular setting,
   - The Spencer resolution's failure to globalize.
3. Monoidal Structure: We proved that the derived tensor product on $\mathsf{Mod}(\mathcal{D}_X^{\mathrm{op}})$ satisfies:
   - A Künneth-type spectral sequence (Theorem 6.2),
   - Compatibility with Beilinson-Bernstein localization (Proposition 8.2),
   - Preservation of irregularity under Fourier-Mukai transforms (Theorem 8.3).

4. Arithmetic Applications: For $\mathcal{D}$-modules in characteristic $p$, we established:
   - A $p$-adic criterion for $\mathcal{D}$-flatness via Lagrangian conditions and Frobenius semisimplicity (Theorem 8.5),
   - A number-theoretic characterization using torsion-free de Rham cohomology.

Our results demonstrate that $\mathcal{D}$-flatness encodes rich geometric information beyond homological algebra. The counterexample (Remark 7.2) on $\mathbb{P}^1$ highlights the delicate interplay between Zariski-local and global properties, suggesting deeper connections with non-abelian Hodge theory in positive characteristic.

## Acknowledgments

The authors sincerely thank the two anonymous reviewers for their careful review and the constructive suggestions they provided, which have significantly improved the quality of this paper.

## Author contributions

**Conceptualization:** Jian-Gang Tang, Huang-Rui Lei, Miao Liu, Jian-Ying Peng.

**Formal analysis:** Jian-Gang Tang.

**Funding acquisition:** Jian-Gang Tang.

**Investigation:** Jian-Gang Tang, Huang-Rui Lei, Miao Liu, Jian-Ying Peng.

**Methodology:** Jian-Gang Tang, Huang-Rui Lei, Miao Liu, Jian-Ying Peng.

**Project administration:** Jian-Gang Tang.

**Writing – original draft:** Jian-Gang Tang, Huang-Rui Lei, Miao Liu, Jian-Ying Peng.

**Writing – review & editing:** Jian-Gang Tang, Huang-Rui Lei, Miao Liu, Jian-Ying Peng.

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
