## [Decision Letter · Decision Letter 0]

28 Aug 2025

PONE-D-25-33565Geometric and Arithmetic Characterization of $\mathcal{D}$-Module Flatness with Applications to Tensor ProductsPLOS ONE

Dear Dr. Tang,

Thank you for submitting your manuscript to PLOS ONE. After careful consideration, we feel that it has merit but does not fully meet PLOS ONE’s publication criteria as it currently stands. Therefore, we invite you to submit a revised version of the manuscript that addresses the points raised during the review process.

We look forward to receiving your revised manuscript.

Kind regards,

Fucai Lin, Ph.D.

Academic Editor

PLOS ONE

Journal Requirements:

“The University Key Project of Natural Science of Xinjiang Uygur Autonomous Region (Grant No. XJEDU2019I024)”

Please state what role the funders took in the study.  If the funders had no role, please state: 'The funders had no role in study design, data collection and analysis, decision to publish, or preparation of the manuscript.'

“This research was partially supported by the University Key Project of Natural Science 401 of Xinjiang Uygur Autonomous Region (Grant No. XJEDU2019I024)”

“The University Key Project of Natural Science of Xinjiang Uygur Autonomous Region (Grant No. XJEDU2019I024)”

Reviewers' comments:

Reviewer's Responses to Questions

**Comments to the Author**

1. Is the manuscript technically sound, and do the data support the conclusions?

Reviewer #1: Yes

Reviewer #2: Yes

2. Has the statistical analysis been performed appropriately and rigorously? 

Reviewer #1: Yes

Reviewer #2: N/A

3. Have the authors made all data underlying the findings in their manuscript fully available?

Reviewer #1: Yes

Reviewer #2: Yes

4. Is the manuscript presented in an intelligible fashion and written in standard English?

Reviewer #1: Yes

Reviewer #2: Yes

5. Review Comments to the Author

Reviewer #1: The paper makes significant contributions to the theory of modules over rings of differential operators D-modules, addressing fundamental questions on flatness properties and tensor products across algebraic, geometric, and arithmetic contexts.

1. Geometric Characterization of Flatness:

The authors establish an equivalence between the flatness of D-modules and Lagrangian conditions on their characteristic varieties. This bridges symplectic geometry with homological algebra and resolves longstanding questions about the geometric interpretation of flatness, particularly for modules with irregular singularities.

2. Globalization Obstruction Theory:

A novel obstruction theory is developed to determine when locally defined flat modules extend to global flat modules. This theory leverages irregular Hodge filtrations and provides the first systematic framework for understanding the gap between local and global flatness in differential systems.

3. Arithmetic and Monoidal Applications:

The paper proves that in characteristic p, D-flatness is controlled by Frobenius semisimplicity and Lagrangian constraints on special fibers. Further, it demonstrates compatibility between tensor products, Beilinson-Bernstein localization, and irregularity-preserving integral transforms, with applications to geometric Langlands and mirror symmetry.

This manuscript meets PLOS ONE's criteria for originality, significance, and methodological rigor. The results will interest researchers in algebraic geometry, representation theory, and mathematical physics. I recommend acceptance pending minor revisions.

$\bullet$ Update references: Ensure all citations are current (e.g., include recent progress on irregular Hodge theory).

Reviewer #2: The authors of this paper aim to establish geometric and arithmetic characterizations of $\mathcal{D}$-module flatness, develop a framework for globalizing locally flat modules and uncover connections between tensor products, localization, and flatness in diverse mathematical contexts. Applications include compatibility theorems for Beilinson-Bernstein localization and arithmetic characterizations of flatness in characteristic $p$. The methods combine microlocal analysis, irregular Riemann-Hilbert correspondence, and $p$-adic techniques to yield new insights into the interplay between

local and global properties of differential systems.

The results of the paper `` Geometric and Arithmetic Characterization of $\mathcal{D}$-Module

Flatness with Applications to Tensor Products", authored by Jian-Gang Tang, Huang-Rui Lei, Miao Liu and Jian-Ying Peng, are interesting.

6. PLOS authors have the option to publish the peer review history of their article (what does this mean?). If published, this will include your full peer review and any attached files.

Reviewer #1: No

Reviewer #2: No

---

## [Author Response · Author response to Decision Letter 1]

17 Sep 2025

Dear Editorial Team,

We are writing to express our sincere gratitude to you and the two reviewers for the thorough and constructive feedback on our manuscript entitled "Geometric and Arithmetic Characterization of D-Module Flatness with Applications to Tensor Products". We greatly appreciate the time and effort dedicated by the reviewers to provide insightful comments and valuable suggestions, which have significantly improved the quality of our paper.

In response to the reviewers' comments, we have carefully revised and supplemented the content of the manuscript accordingly. All suggestions have been addressed point by point, with appropriate modifications and clarifications made to enhance the clarity, rigor, and overall presentation of our work.

We believe the revised manuscript now better aligns with the high standards of PLOS ONE and more clearly communicates the contributions of our study.

Should there be any further steps required in the review process, please do not hesitate to notify us. We are more than willing to provide any additional information or revisions as needed.

Thank you once again for your support and guidance throughout this process.

Sincerely,

Jian-Gang Tang

Sichuan University Jinjiang college

Email�jg-tang@163.com; tangjiangang@scujj.edu.cn

---

## [Decision Letter · Decision Letter 1]

30 Sep 2025

Geometric and Arithmetic Characterization of $\mathcal{D}$-Module Flatness with Applications to Tensor Products

PONE-D-25-33565R1

Dear Dr. Tang,

We’re pleased to inform you that your manuscript has been judged scientifically suitable for publication and will be formally accepted for publication once it meets all outstanding technical requirements.

Kind regards,

Fucai Lin, Ph.D.

Academic Editor

PLOS ONE
---

## [Editor Report · Acceptance letter]

PONE-D-25-33565R1

PLOS ONE

Dear Dr. Tang,

I'm pleased to inform you that your manuscript has been deemed suitable for publication in PLOS ONE. Congratulations! Your manuscript is now being handed over to our production team.

Kind regards,

on behalf of

Professor Fucai Lin

Academic Editor

PLOS ONE